# Deep Exploration via Bootstrapped DQN

**Ian Osband**[1,2]**, Charles Blundell**[2]**, Alexander Pritzel**[2]**, Benjamin Van Roy**[1]
[1]Stanford University, [2]Google DeepMind
{iosband, cblundell, apritzel}@google.com, bvr@stanford.edu

## Abstract

Efficient exploration remains a major challenge for reinforcement learning (RL). Common dithering strategies for exploration, such as $\epsilon$-greedy, do not carry out temporally-extended (or deep) exploration; this can lead to exponentially larger data requirements. However, most algorithms for statistically efficient RL are not computationally tractable in complex environments. Randomized value functions offer a promising approach to efficient exploration with generalization, but existing algorithms are not compatible with nonlinearly parameterized value functions. As a first step towards addressing such contexts we develop *bootstrapped DQN*. We demonstrate that bootstrapped DQN can combine deep exploration with deep neural networks for exponentially faster learning than any dithering strategy. In the Arcade Learning Environment bootstrapped DQN substantially improves learning speed and cumulative performance across most games.

## 1 Introduction

We study the reinforcement learning (RL) problem where an agent interacts with an unknown environment. The agent takes a sequence of actions in order to maximize cumulative rewards. Unlike standard planning problems, an RL agent does not begin with perfect knowledge of the environment, but learns through experience. This leads to a fundamental trade-off of exploration versus exploitation; the agent may improve its future rewards by exploring poorly understood states and actions, but this may require sacrificing immediate rewards. To learn efficiently an agent should explore only when there are valuable learning opportunities. Further, since any action may have long term consequences, the agent should reason about the informational value of possible observation sequences. Without this sort of temporally extended (deep) exploration, learning times can worsen by an exponential factor.

The theoretical RL literature offers a variety of provably-efficient approaches to deep exploration [9]. However, most of these are designed for Markov decision processes (MDPs) with small finite state spaces, while others require solving computationally intractable planning tasks [8]. These algorithms are not practical in complex environments where an agent must generalize to operate effectively. For this reason, large-scale applications of RL have relied upon statistically inefficient strategies for exploration [12] or even no exploration at all [23]. We review related literature in more detail in Section 4.

Common dithering strategies, such as $\epsilon$-greedy, approximate the value of an action by a single number. Most of the time they pick the action with the highest estimate, but sometimes they choose another action at random. In this paper, we consider an alternative approach to efficient exploration inspired by Thompson sampling. These algorithms have some notion of uncertainty and instead maintain a *distribution* over possible values. They explore by randomly select a policy according to the probability it is the optimal policy. Recent work has shown that randomized value functions can implement something similar to Thompson sampling without the need for an intractable exact posterior update. However, this work is restricted to linearly-parameterized value functions [16]. We present a natural

extension of this approach that enables use of complex non-linear generalization methods such as deep neural networks. We show that the bootstrap with random initialization can produce reasonable uncertainty estimates for neural networks at low computational cost. Bootstrapped DQN leverages these uncertainty estimates for efficient (and deep) exploration. We demonstrate that these benefits can extend to large scale problems that are not designed to highlight deep exploration. Bootstrapped DQN substantially reduces learning times and improves performance across most games. This algorithm is computationally efficient and parallelizable; on a single machine our implementation runs roughly 20% slower than DQN.

## 2   Uncertainty for neural networks

Deep neural networks (DNN) represent the state of the art in many supervised and re-inforcement learning domains [12]. We want an exploration strategy that is statistically computationally efficient together with a DNN representation of the value function. To explore efficiently, the first step to quantify uncertainty in value estimates so that the agent can judge potential benefits of exploratory actions. The neural network literature presents a sizable body of work on uncertainty quantification founded on parametric Bayesian inference [3, 7]. We actually found the simple non-parametric bootstrap with random initialization [5] more effective in our experiments, but the main ideas of this paper would apply with any other approach to uncertainty in DNNs.

The bootstrap principle is to approximate a population distribution by a sample distribution [6]. In its most common form, the bootstrap takes as input a data set $D$ and an estimator $\psi$. To generate a sample from the bootstrapped distribution, a data set $\tilde{D}$ of cardinality equal to that of $D$ is sampled uniformly with replacement from $D$. The bootstrap sample estimate is then taken to be $\psi(\tilde{D})$. The bootstrap is widely hailed as a great advance of 20th century applied statistics and even comes with theoretical guarantees [2]. In Figure 1a we present an efficient and scalable method for generating bootstrap samples from a large and deep neural network. The network consists of a shared architecture with $K$ bootstrapped "heads" branching off independently. Each head is trained only on its bootstrapped sub-sample of the data and represents a single bootstrap sample $\psi(\tilde{D})$. The shared network learns a joint feature representation across all the data, which can provide significant computational advantages at the cost of lower diversity between heads. This type of bootstrap can be trained efficiently in a single forward/backward pass; it can be thought of as a data-dependent dropout, where the dropout mask for each head is fixed for each data point [19].

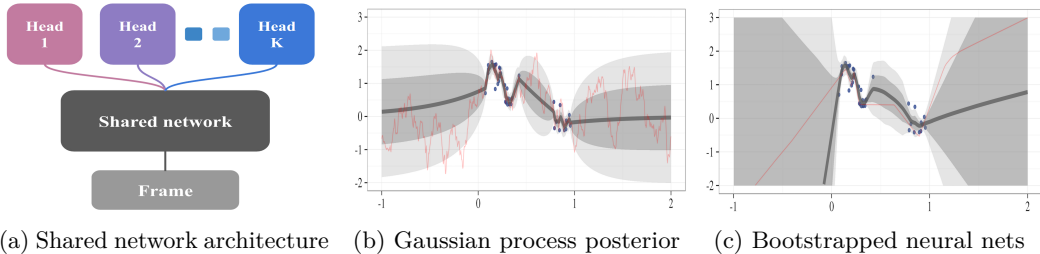

(a) Shared network architecture   (b) Gaussian process posterior   (c) Bootstrapped neural nets

Figure 1: Bootstrapped neural nets can produce reasonable posterior estimates for regression.

Figure 1 presents an example of uncertainty estimates from bootstrapped neural networks on a regression task with noisy data. We trained a fully-connected 2-layer neural networks with 50 rectified linear units (ReLU) in each layer on 50 bootstrapped samples from the data. As is standard, we initialize these networks with random parameter values, this induces an important initial diversity in the models. We were unable to generate effective uncertainty estimates for this problem using the dropout approach in prior literature [7]. Further details are provided in Appendix A.

## 3   Bootstrapped DQN

For a policy $\pi$ we define the value of an action $a$ in state $s$ $Q^\pi(s,a) := \mathbb{E}_{s,a,\pi}\left[\sum_{t=1}^{\infty} \gamma^t r_t\right]$, where $\gamma \in (0,1)$ is a discount factor that balances immediate versus future rewards $r_t$. This expectation indicates that the initial state is $s$, the initial action is $a$, and thereafter actions

are selected by the policy $\pi$. The optimal value is $Q^*(s, a) := \max_\pi Q^\pi(s, a)$. To scale to large problems, we learn a parameterized estimate of the Q-value function $Q(s, a; \theta)$ rather than a tabular encoding. We use a neural network to estimate this value.

The Q-learning update from state $s_t$, action $a_t$, reward $r_t$ and new state $s_{t+1}$ is given by

$$\theta_{t+1} \leftarrow \theta_t + \alpha(y_t^Q - Q(s_t, a_t; \theta_t))\nabla_\theta Q(s_t, a_t; \theta_t) \tag{1}$$

where $\alpha$ is the scalar learning rate and $y_t^Q$ is the target value $r_t + \gamma \max_a Q(s_{t+1}, a; \theta^-)$. $\theta^-$ are target network parameters fixed $\theta^- = \theta_t$.

Several important modifications to the Q-learning update improve stability for DQN [12]. First the algorithm learns from sampled transitions from an experience buffer, rather than learning fully online. Second the algorithm uses a target network with parameters $\theta^-$ that are copied from the learning network $\theta^- \leftarrow \theta_t$ only every $\tau$ time steps and then kept fixed in between updates. Double DQN [25] modifies the target $y_t^Q$ and helps further[1]:

$$y_t^Q \leftarrow r_t + \gamma \max_a Q\big(s_{t+1}, \arg\max_a Q(s_{t+1}, a; \theta_t); \theta^-\big). \tag{2}$$

Bootstrapped DQN modifies DQN to approximate a *distribution* over Q-values via the bootstrap. At the start of each episode, bootstrapped DQN samples a single Q-value function from its approximate posterior. The agent then follows the policy which is optimal for that *sample* for the duration of the episode. This is a natural adaptation of the Thompson sampling heuristic to RL that allows for temporally extended (or deep) exploration [21, 13].

We implement this algorithm efficiently by building up $K \in \mathbb{N}$ bootstrapped estimates of the Q-value function in parallel as in Figure 1a. Importantly, each one of these value function function heads $Q_k(s, a; \theta)$ is trained against its own target network $Q_k(s, a; \theta^-)$. This means that each $Q_1, .., Q_K$ provide a temporally extended (and consistent) estimate of the value uncertainty via TD estimates. In order to keep track of which data belongs to which bootstrap head we store flags $w_1, .., w_K \in \{0, 1\}$ indicating which heads are privy to which data. We approximate a bootstrap sample by selecting $k \in \{1, .., K\}$ uniformly at random and following $Q_k$ for the duration of that episode. We present a detailed algorithm for our implementation of bootstrapped DQN in Appendix B.

## 4  Related work

The observation that temporally extended exploration is necessary for efficient reinforcement learning is not new. For any prior distribution over MDPs, the optimal exploration strategy is available through dynamic programming in the Bayesian belief state space. However, the exact solution is intractable even for very simple systems[8]. Many successful RL applications focus on generalization and planning but address exploration only via inefficient exploration [12] or even none at all [23]. However, such exploration strategies can be highly inefficient.

Many exploration strategies are guided by the principle of "optimism in the face of uncertainty" (OFU). These algorithms add an exploration bonus to values of state-action pairs that may lead to useful learning and select actions to maximize these adjusted values. This approach was first proposed for finite-armed bandits [11], but the principle has been extended successfully across bandits with generalization and tabular RL [9]. Except for particular deterministic contexts [27], OFU methods that lead to efficient RL in complex domains have been computationally intractable. The work of [20] aims to add an effective bonus through a variation of DQN. The resulting algorithm relies on a large number of hand-tuned parameters and is only suitable for application to deterministic problems. We compare our results on Atari to theirs in Appendix D and find that bootstrapped DQN offers a significant improvement over previous methods.

Perhaps the oldest heuristic for balancing exploration with exploitation is given by Thompson sampling [24]. This bandit algorithm takes a single sample from the posterior at every time step and chooses the action which is optimal for that time step. To apply the Thompson sampling principle to RL, an agent should sample a value function from its posterior. Naive applications of Thompson sampling to RL which resample every timestep can be extremely

inefficient. The agent must also commit to this sample for several time steps in order to achieve deep exploration [21, 8]. The algorithm PSRL does exactly this, with state of the art guarantees [13, 14]. However, this algorithm still requires solving a single known MDP, which will usually be intractable for large systems.

Our new algorithm, bootstrapped DQN, approximates this approach to exploration via randomized value functions sampled from an approximate posterior. Recently, authors have proposed the RLSVI algorithm which accomplishes this for linearly parameterized value functions. Surprisingly, RLSVI recovers state of the art guarantees in the setting with tabular basis functions, but its performance is crucially dependent upon a suitable linear representation of the value function [16]. We extend these ideas to produce an algorithm that can simultaneously perform generalization and exploration with a flexible nonlinear value function representation. Our method is simple, general and compatible with almost all advances in deep RL at low computational cost and with few tuning parameters.

## 5  Deep Exploration

Uncertainty estimates allow an agent to direct its exploration at potentially informative states and actions. In bandits, this choice of directed exploration rather than dithering generally categorizes efficient algorithms. The story in RL is not as simple, directed exploration is not enough to guarantee efficiency; the exploration must also be deep. Deep exploration means exploration which is directed over multiple time steps; it can also be called "planning to learn" or "far-sighted" exploration. Unlike bandit problems, which balance actions which are immediately rewarding or immediately informative, RL settings require planning over several time steps [10]. For exploitation, this means that an efficient agent must consider the future rewards over several time steps and not simply the myopic rewards. In exactly the same way, efficient exploration may require taking actions which are neither immediately rewarding, nor immediately informative.

To illustrate this distinction, consider a simple deterministic chain $\{s_{-3}, .., s_{+3}\}$ with three step horizon starting from state $s_0$. This MDP is known to the agent a priori, with deterministic actions "left" and "right". All states have zero reward, except for the leftmost state $s_{-3}$ which has known reward $\epsilon > 0$ and the rightmost state $s_3$ which is unknown. In order to reach either a rewarding state or an informative state within three steps from $s_0$ the agent must plan a consistent strategy over several time steps. Figure 2 depicts the planning and look ahead trees for several algorithmic approaches in this example MDP. The action "left" is gray, the action "right" is black. Rewarding states are depicted as red, informative states as blue. Dashed lines indicate that the agent can plan ahead for either rewards or information. Unlike bandit algorithms, an RL agent can plan to exploit future rewards. Only an RL agent with deep exploration can plan to learn.

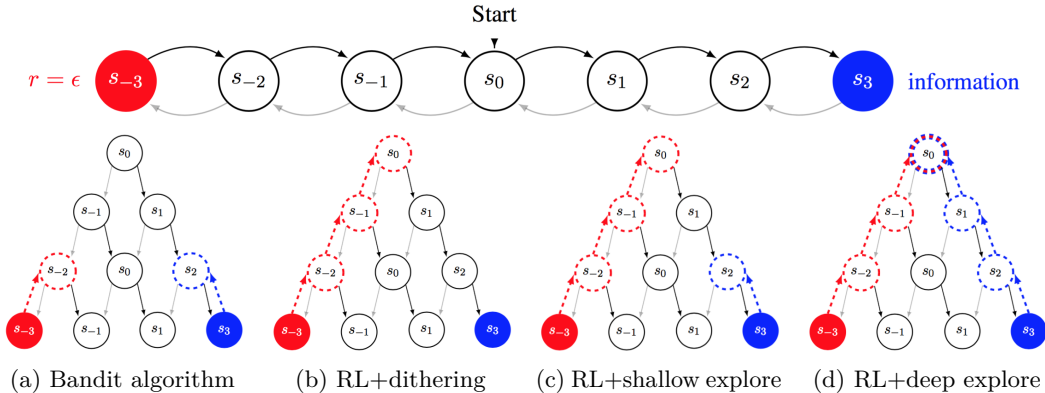

(a) Bandit algorithm    (b) RL+dithering    (c) RL+shallow explore    (d) RL+deep explore

Figure 2: Planning, learning and exploration in RL.

## 5.1 Testing for deep exploration

We now present a series of didactic computational experiments designed to highlight the need for deep exploration. These environments can be described by chains of length $N > 3$ in Figure 3. Each episode of interaction lasts $N + 9$ steps after which point the agent resets to the initial state $s_2$. These are toy problems intended to be expository rather than entirely realistic. Balancing a well known and mildly successful strategy versus an unknown, but potentially more rewarding, approach can emerge in many practical applications.

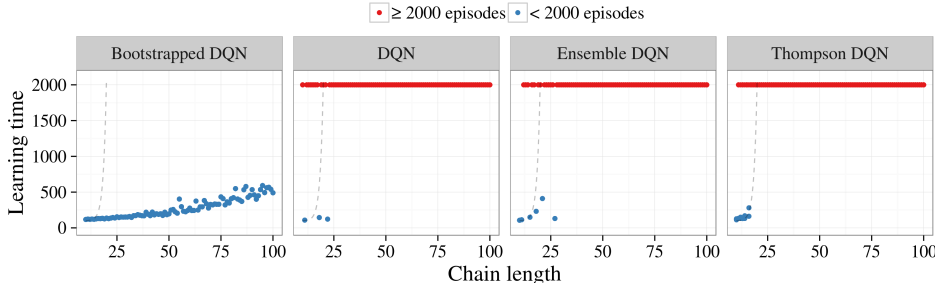

Figure 3: Scalable environments that requires deep exploration.

These environments may be described by a finite tabular MDP. However, we consider algorithms which interact with the MDP only through raw pixel features. We consider two feature mappings $\phi_{1\text{hot}}(s_t) := (\mathbb{1}\{x = s_t\})$ and $\phi_{\text{therm}}(s_t) := (\mathbb{1}\{x \leq s_t\})$ in $\{0, 1\}^N$. We present results for $\phi_{\text{therm}}$, which worked better for all DQN variants due to better generalization, but the difference was relatively small - see Appendix C. Thompson DQN is the same as bootstrapped DQN, but resamples every timestep. Ensemble DQN uses the same architecture as bootstrapped DQN, but with an ensemble policy.

We say that the algorithm has successfully learned the optimal policy when it has successfully completed one hundred episodes with optimal reward of 10. For each chain length, we ran each learning algorithm for 2000 episodes across three seeds. We plot the median time to learn in Figure 4, together with a conservative lower bound of $99 + 2^{N-11}$ on the expected time to learn for any shallow exploration strategy [16]. Only bootstrapped DQN demonstrates a graceful scaling to long chains which require deep exploration.

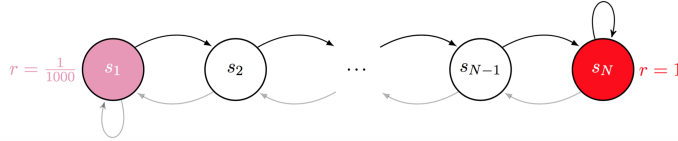

Figure 4: Only Bootstrapped DQN demonstrates deep exploration.

## 5.2 How does bootstrapped DQN drive deep exploration?

Bootstrapped DQN explores in a manner similar to the provably-efficient algorithm PSRL [13] but it uses a bootstrapped neural network to approximate a posterior sample for the value. Unlike PSRL, bootstrapped DQN directly samples a value function and so does not require further planning steps. This algorithm is similar to RLSVI, which is also provably-efficient [16], but with a neural network instead of linear value function and bootstrap instead of Gaussian sampling. The analysis for the linear setting suggests that this nonlinear approach will work well so long as the distribution $\{Q^1, .., Q^K\}$ remains *stochastically optimistic* [16], or at least as spread out as the "correct" posterior.

Bootstrapped DQN relies upon random initialization of the network weights as a prior to induce diversity. Surprisingly, we found this initial diversity was enough to maintain diverse generalization to new and unseen states for large and deep neural networks. This is effective for our experimental setting, but will not work in all situations. In general it may be necessary to maintain some more rigorous notion of "prior", potentially through the use of artificial prior data to maintain diversity [15]. One potential explanation for the efficacy of simple random initialization is that unlike supervised learning or bandits, where all networks fit the same data, each of our $Q^k$ heads has a unique target network. This, together with stochastic minibatch and flexible nonlinear representations, means that even small differences at initialization may become *bigger* as they refit to unique TD errors.

Bootstrapped DQN does *not* require that any single network $Q^k$ is initialized to the correct policy of "right" at every step, which would be exponentially unlikely for large chains $N$. For the algorithm to be successful in this example we only require that the networks generalize in a diverse way to the actions they have never chosen in the states they have not visited very often. Imagine that, in the example above, the network has made it as far as state $\tilde{N} < N$, but never observed the action right $a = 2$. As long as one head $k$ imagines $Q(\tilde{N}, 2) > Q(\tilde{N}, 2)$ then TD bootstrapping can propagate this signal back to $s = 1$ through the target network to drive deep exploration. The expected time for these estimates at $n$ to propagate to at least one head grows gracefully in $n$, even for relatively small $K$, as our experiments show. We expand upon this intuition with a video designed to highlight *how* bootstrapped DQN demonstrates deep exploration `https://youtu.be/e3KuV_d0EMk`. We present further evaluation on a difficult stochastic MDP in Appendix C.

# 6 Arcade Learning Environment

We now evaluate our algorithm across 49 Atari games on the Arcade Learning Environment [1]. Importantly, and unlike the experiments in Section 5, these domains are not specifically designed to showcase our algorithm. In fact, many Atari games are structured so that small rewards always indicate part of an optimal policy. This may be crucial for the strong performance observed by dithering strategies[2]. We find that exploration via bootstrapped DQN produces significant gains versus $\epsilon$-greedy in this setting. Bootstrapped DQN reaches peak performance roughly similar to DQN. However, our improved exploration mean we reach human performance on average 30% faster across all games. This translates to significantly improved cumulative rewards through learning.

We follow the setup of [25] for our network architecture and benchmark our performance against their algorithm. Our network structure is identical to the convolutional structure of DQN [12] except we split 10 separate bootstrap heads after the convolutional layer as per Figure 1a. Recently, several authors have provided architectural and algorithmic improvements to DDQN [26, 18]. We do not compare our results to these since their advances are orthogonal to our concern and could easily be incorporated to our bootstrapped DQN design. Full details of our experimental set up are available in Appendix D.

## 6.1 Implementing bootstrapped DQN at scale

We now examine how to generate online bootstrap samples for DQN in a computationally efficient manner. We focus on three key questions: how many heads do we need, how should we pass gradients to the shared network and how should we bootstrap data online? We make significant compromises in order to maintain computational cost comparable to DQN.

Figure 5a presents the cumulative reward of bootstrapped DQN on the game Breakout, for different number of heads $K$. More heads leads to faster learning, but even a small number of heads captures most of the benefits of bootstrapped DQN. We choose $K = 10$.

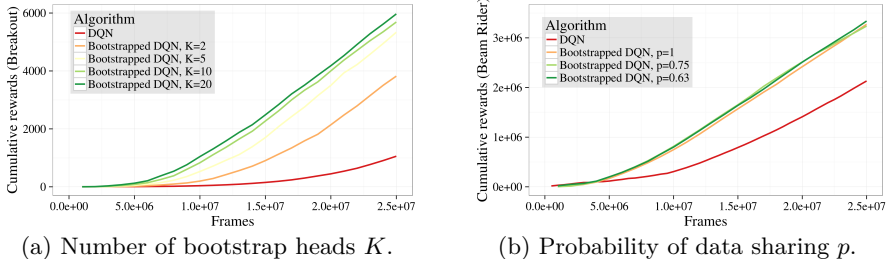

(a) Number of bootstrap heads $K$.  (b) Probability of data sharing $p$.

Figure 5: Examining the sensitivities of bootstrapped DQN.

The shared network architecture allows us to train this combined network via backpropagation. Feeding $K$ network heads to the shared convolutional network effectively increases the learning rate for this portion of the network. In some games, this leads to premature and sub-optimal convergence. We found the best final scores by normalizing the gradients by $1/K$, but this also leads to slower early learning. See Appendix D for more details.

To implement an online bootstrap we use an independent Bernoulli mask $w_1,..,w_K \sim \mathrm{Ber}(p)$ for each head in each episode[3]. These flags are stored in the memory replay buffer and identify which heads are trained on which data. However, when trained using a shared minibatch the algorithm will also require an effective $1/p$ more iterations; this is undesirable computationally. Surprisingly, we found the algorithm performed similarly irrespective of $p$ and all outperformed DQN, as shown in Figure 5b. This is strange and we discuss this phenomenon in Appendix D. However, in light of this empirical observation for Atari, we chose $p=1$ to save on minibatch passes. As a result bootstrapped DQN runs at similar computational speed to vanilla DQN on identical hardware[4].

## 6.2  Efficient exploration in Atari

We find that Bootstrapped DQN drives efficient exploration in several Atari games. For the same amount of game experience, bootstrapped DQN generally outperforms DQN with $\epsilon$-greedy exploration. Figure 6 demonstrates this effect for a diverse selection of games.

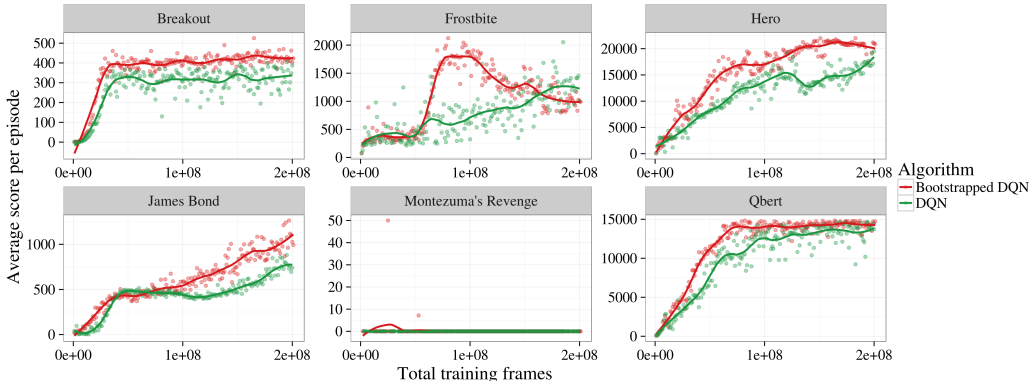

Figure 6: Bootstrapped DQN drives more efficient exploration.

On games where DQN performs well, bootstrapped DQN typically performs better. Bootstrapped DQN does not reach human performance on Amidar (DQN does) but does on Beam Rider and Battle Zone (DQN does not). To summarize this improvement in learning time we consider the number of frames required to reach human performance. If bootstrapped DQN reaches human performance in $1/x$ frames of DQN we say it has improved by $x$. Figure 7 shows that Bootstrapped DQN typically reaches human performance significantly faster.

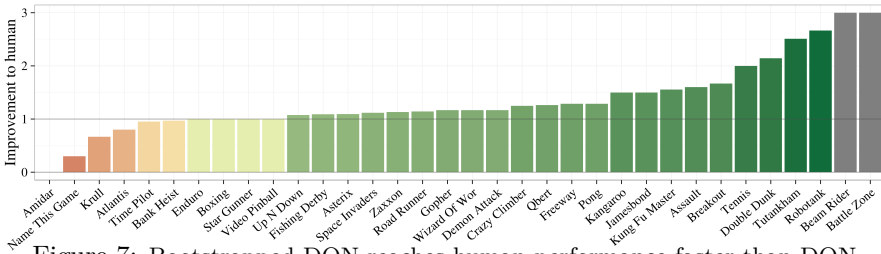

Figure 7: Bootstrapped DQN reaches human performance faster than DQN.

On most games where DQN does not reach human performance, bootstrapped DQN does not solve the problem by itself. On some challenging Atari games where deep exploration is conjectured to be important [25] our results are not entirely successful, but still promising. In Frostbite, bootstrapped DQN reaches the second level much faster than DQN but network instabilities cause the performance to crash. In Montezuma's Revenge, bootstrapped DQN reaches the first key after 20m frames (DQN never observes a reward even after 200m frames) but does not properly learn from this experience[5]. Our results suggest that improved exploration may help to solve these remaining games, but also highlight the importance of other problems like network instability, reward clipping and temporally extended rewards.

## 6.3 Overall performance

Bootstrapped DQN is able to learn much faster than DQN. Figure 8 shows that bootstrapped DQN also improves upon the final score across most games. However, the real benefits to *efficient* exploration mean that bootstrapped DQN outperforms DQN by orders of magnitude in terms of the *cumulative* rewards through learning (Figure 9. In both figures we normalize performance relative to a fully random policy. The most similar work to ours presents several other approaches to improved exploration in Atari [20] they optimize for AUC-20, a normalized version of the cumulative returns after 20m frames. According to their metric, averaged across the 14 games they consider, we improve upon both base DQN (0.29) and their best method (0.37) to obtain 0.62 via bootstrapped DQN. We present these results together with results tables across all 49 games in Appendix D.4.

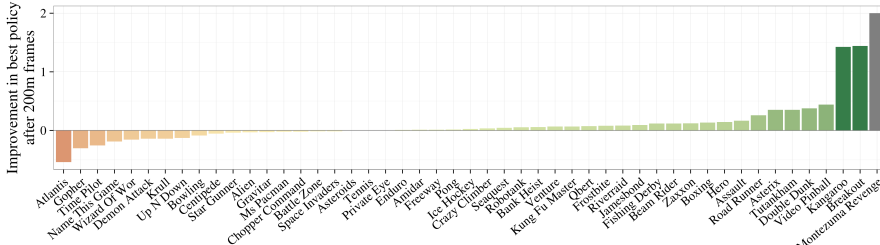

Figure 8: Bootstrapped DQN typically improves upon the best policy.

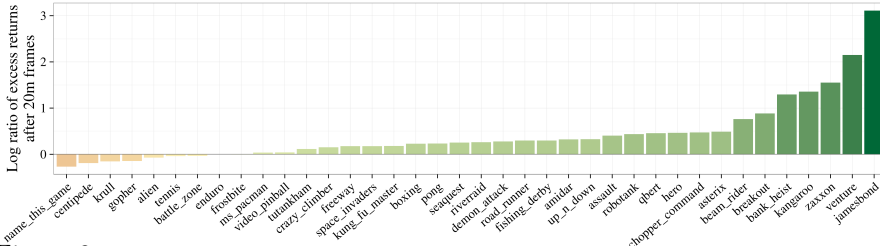

Figure 9: Bootstrapped DQN improves cumulative rewards by orders of magnitude.

## 6.4 Visualizing bootstrapped DQN

We now present some more insight to how bootstrapped DQN drives deep exploration in Atari. In each game, although each head $Q^1, .., Q^{10}$ learns a high scoring policy, the policies they find are quite distinct. In the video `https://youtu.be/Zm2KoT82O_M` we show the evolution of these policies simultaneously for several games. Although each head performs well, they each follow a unique policy. By contrast, $\epsilon$-greedy strategies are almost indistinguishable for small values of $\epsilon$ and totally ineffectual for larger values. We believe that this deep exploration is key to improved learning, since diverse experiences allow for better generalization.

Disregarding exploration, bootstrapped DQN may be beneficial as a purely exploitative policy. We can combine all the heads into a single ensemble policy, for example by choosing the action with the most votes across heads. This approach might have several benefits. First, we find that the ensemble policy can often outperform any individual policy. Second, the distribution of votes across heads to give a measure of the uncertainty in the optimal policy. Unlike vanilla DQN, bootstrapped DQN can know what it doesn't know. In an application where executing a poorly-understood action is dangerous this could be crucial. In the video `https://youtu.be/0jvEcC5JvGY` we visualize this ensemble policy across several games. We find that the uncertainty in this policy is surprisingly interpretable: all heads agree at clearly crucial decision points, but remain diverse at other less important steps.

## 7 Closing remarks

In this paper we present bootstrapped DQN as an algorithm for efficient reinforcement learning in complex environments. We demonstrate that the bootstrap can produce useful uncertainty estimates for deep neural networks. Bootstrapped DQN is computationally tractable and also naturally scalable to massive parallel systems. We believe that, beyond our specific implementation, randomized value functions represent a promising alternative to dithering for exploration. Bootstrapped DQN practically combines efficient generalization with exploration for complex nonlinear value functions.

## Footnotes

[1]In this paper we use the DDQN update for all DQN variants unless explicitly stated.

[2]By contrast, imagine that the agent received a small immediate reward for dying; dithering strategies would be hopeless at solving this problem, just like Section 5.

[3]$p=0.5$ is double-or-nothing bootstrap [17], $p=1$ is ensemble with no bootstrapping at all.

[4]Our implementation $K=10$, $p=1$ ran with less than a 20% increase on wall-time versus DQN.

[5]An improved training method, such as prioritized replay [18] may help solve this problem.

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
