[Supplementary Material · boot_dqn_nips_appendix.pdf]

# APPENDICES

## A Uncertainty for neural networks

In this appendix we discuss some of the experimental setup to qualitatively evaluate uncertainty methods for deep neural networks. To do this, we generated twenty noisy regression pairs $x_i, y_i$ with:

$$y_i = x_i + sin(\alpha(x_i + w_i)) + sin(\beta(x_i + w_i)) + w_i$$

where $x_i$ are drawn uniformly from $(0, 0.6) \cup (0.8, 1)$ and $w_i \sim N(\mu = 0, \sigma^2 = 0.03^2)$. We set $\alpha = 4$ and $\beta = 13$. None of these numerical choices were important except to represent a highly nonlinear function with lots of noise and several clear regions where we should be uncertain. We present the regression data together with an indication of the generating distribution in Figure 10.

Figure 10: Underlying generating distribution. All our algorithms receive the same blue data. Pink points represent other samples, the mean function is shown in black.

Interestingly, we did not find that using dropout produced satisfying confidence intervals for this task. We present one example of this dropout posterior estimate in Figure 11a.

(a) Dropout gives strange uncertainty estimates.

(b) Screenshot from accompanying web demo to [7]. Dropout converges with high certainty to the mean value.

Figure 11: Comparing the bootstrap to dropout uncertainty for neural nets.

These results are unsatisfactory for several reasons. First, the network extrapolates the mean posterior far outside the range of any actual data for $x = 0.75$. We believe this is because dropout only perturbs locally from a single neural network fit, unlike bootstrap. Second, the posterior samples from the dropout approximation are very spiky and do not look like any sensible posterior sample. Third, the network collapses to almost zero uncertainty in regions with data.

We spent some time altering our dropout scheme to fix this effect, which might be undesirable for stochastic domains and we believed might be an artefact of our implementation. However,

after further thought we believe this to be an effect which you would expect for dropout posterior approximations. In Figure 11b we present a didactic example taken from the author's website [7].

On the right hand side of the plot we generate noisy data with wildly different values. Training a neural network using MSE criterion means that the network will surely converge to the mean of the noisy data. Any dropout samples remain highly concentrated around this mean. By contrast, bootstrapped neural networks may include different subsets of this noisy data and so may produce a more intuitive uncertainty estimates for our settings. Note this isn't necessarily a failure of dropout to approximate a Gaussian process posterior, but this artefact could be shared by any homoskedastic posterior. The authors of [7] propose a heteroskedastic variant which can help, but does not address the fundamental issue that for large networks trained to convergence all dropout samples may converge to every single datapoint... even the outliers.

This observation is key to another, more essential flaw with naive dropout as a proxy for uncertainty in deep learning. Previous analysis argues that dropout acts as a variational approximation to the posterior distribution of outcomes $p(y)$ [7], however it does not distinguish the *risk* (inherent stochasticity in a model) from the *uncertainty* (the confusion over which model parameters apply). For Thompson sampling, it is important to sample only over the uncertainty over the expected returns and not over realizations of the stochastic rewards. Bootstrap and dropout are not all that different when viewed in a certain light. Bootstrap can be regarded as a data-dependent dropout where each datapoint uniquely determines a mask (which we call the bootstrap mask). Our implementation considers bootstrap masks which completely share parameters in the convnet and are completely distinct in the heads, but we might consider other more general bootstrap masks. Exploring variations of these ideas is an interesting topic for future research.

In this paper we focus on the bootstrap approach to uncertainty for neural networks. We like its simplicity, connections to established statistical methodology and empirical good performance. However, the key insights of this paper is the use of deep exploration via randomized value functions. This is compatible with any approximate posterior estimator for deep neural networks. We believe that this area of uncertainty estimates for neural networks remains an important area of research in its own right.

Bootstrapped uncertainty estimates for the Q-value functions have another crucial advantage over dropout which does not appear in the supervised problem. Unlike random dropout masks trained against random target networks, our implementation of bootstrap DQN trains against its own *temporally consistent* target network. This means that our bootstrap estimates (in the sense of [5]), are able to "bootstrap" (in the TD sense of [22]) on their own estimates of the long run value. This is important to quantify the long run uncertainty over Q and drive deep exploration.

## B Bootstrapped DQN implementation

Algorithm 1 gives a full description of Bootstrapped DQN. It captures two modes of operation where either $k$ neural networks are used to estimate the $Q_k$-value functions, or where one neural network with $k$ heads is used to estimate $k$ Q-value functions. In both cases, as this is largely a parameterization issue, we denote the value function networks as $Q$, where $Q_k$ is output of the $k$th network or the $k$th head.

A core idea to the full bootstrapped DQN algorithm is the bootstrap mask $m_t$. The mask $m_t$ decides, for each value function $Q_k$, whether or not it should train upon the experience generated at step $t$. In its simplest form $m_t$ is a binary vector of length $K$, masking out or including each value function for training on that time step of experience (i.e., should it receive gradients from the corresponding $(s_t, a_t, r_{t+1}, s_{t+1}, m_t)$ tuple). The masking distribution $M$ is responsible for generating each $m_t$. For example, when $M$ yields $m_t$ whose components are independently drawn from a Bernoulli distribution with parameter 0.5 then this corresponds to the double-or-nothing bootstrap [17]. On the other hand, if $M$ yields a mask $m_t$ with all ones, then the algorithm reduces to an ensemble method. Poisson masks $M_t[k] \sim \text{Poi}(1)$ provides the most natural parallel with the standard non-parametric

bootstrap since $\text{Bin}(N, 1/N) \to \text{Poi}(1)$ as $N \to \infty$. Exponential masks $M_t[k] \sim \text{Exp}(1)$ closely resemble the standard Bayesian nonparametric posterior of a Dirichlet process [15].

---

**Algorithm 1** Bootstrapped DQN

---

1: **Input:** Value function networks $Q$ with $K$ outputs $\{Q_k\}_{k=1}^K$. Masking distribution $M$.
2: Let $B$ be a replay buffer storing experience for training.
3: **for** each episode **do**
4:      Obtain initial state from environment $s_0$
5:      Pick a value function to act using $k \sim \text{Uniform}\{1, \ldots, K\}$
6:      **for** step $t = 1, \ldots$ until end of episode **do**
7:          Pick an action according to $a_t \in \arg\max_a Q_k(s_t, a)$
8:          Receive state $s_{t+1}$ and reward $r_t$ from environment, having taking action $a_t$
9:          Sample bootstrap mask $m_t \sim M$
10:         Add $(s_t, a_t, r_{t+1}, s_{t+1}, m_t)$ to replay buffer $B$
11:      **end for**
12: **end for**

---

Periodically, the replay buffer is played back to update the parameters of the value function network $Q$. The gradients of the $k$th value function $Q_k$ for the $t$th tuple in the replay buffer $B$, $g_t^k$ is:

$$g_t^k = m_t^k (y_t^Q - Q_k(s_t, a_t; \theta)) \nabla_\theta Q_k(s_t, a_t; \theta) \tag{3}$$

where $y_t^Q$ is given by (2). Note that the mask $m_t^k$ modulates the gradient, giving rise to the bootstrap behavior.

## C Experiments for deep exploration

### C.1 Bootstrap methodology

A naive implementation of bootstrapped DQN builds up $K$ complete networks with $K$ distinct memory buffers. This method is parallelizable up to many machines, however we wanted to produce an algorithm that was efficient even on a single machine. To do this, we implemented the bootstrap heads in a single larger network, like Figure 1a but without any shared network. We implement bootstrap by masking each episode of data according to $w_1, .., w_K \sim \text{Ber}(p)$.

Figure 12: Bootstrapped DQN performs well even with small number of bootstrap heads $K$ or high probability of sharing $p$.

In Figure 12 we demonstrate that bootstrapped DQN can implement deep exploration even with relatively small values of $K$. However, the results are more robust and scalable with larger $K$. We run our experiments on the example from Figure 3. Surprisingly, this method is even effective with $p = 1$ and complete data sharing between heads. This degenerate full sharing of information turns out to be remarkably efficient for training large and deep neural networks. We discuss this phenomenon more in Appendix D.

Generating good estimates for uncertainty is not enough for efficient exploration. In Figure 13 we see that other methods trained with the same network architecture are totally ineffective at implementing deep exploration. The $\epsilon$-greedy policy follows just one $Q$-value estimate. We allow this policy to be evaluated without dithering. The ensemble policy is trained exactly as per bootstrapped DQN except at each stage the algorithm follows the policy which is majority vote of the bootstrap heads. Thompson sampling is the same as bootstrapped DQN except a new head is sampled every timestep, rather than every episode.

Figure 13: Shallow exploration methods do not work.

We can see that only bootstrapped DQN demonstrates efficient and deep exploration in this domain.

## C.2  A difficult stochastic MDP

Figure 4 shows that bootstrapped DQN can implement effective (and deep) exploration where similar deep RL architectures fail. However, since the underlying system is a small and finite MDP there may be several other simpler strategies which would also solve this problem. We will now consider a difficult variant of this chain system with significant stochastic noise in transitions as depicted in Figure 14. Action "left" deterministically moves the agent left, but action "right" is only successful 50% of the time and otherwise also moves left. The agent interacts with the MDP in episodes of length 15 and begins each episode at $s_1$. Once again the optimal policy is to head right.

Figure 14: A stochastic MDP that requires deep exploration.

Bootstrapped DQN is unique amongst scalable approaches to efficient exploration with deep RL in stochastic domains. For benchmark performance we implement three algorithms which, unlike bootstrapped DQN, will receive the true tabular representation for the MDP. These algorithms are based on three state of the art approaches to exploration via dithering ($\epsilon$-greedy), optimism [9] and posterior sampling [13]. We discuss the choice of these benchmarks in Appendix C.

In Figure 15a we present the empirical regret of each algorithm averaged over 10 seeds over the first two thousand episodes. The empirical regret is the cumulative difference between the expected rewards of the optimal policy and the realized rewards of each algorithm. We find that bootstrapped DQN achieves similar performance to state of the art efficient exploration schemes such as PSRL even without prior knowledge of the tabular MDP structure and in noisy environments.

Most telling is how much better bootstrapped DQN does than the state of the art optimistic algorithm UCRL2. Although Figure 15a seems to suggest UCRL2 incurs linear regret,

(a) Bootstrapped DQN matches efficient tabular RL.

(b) The regret bounds for UCRL2 are near-optimal in $\tilde{O}(\cdot)$, but they are still not very practical.

Figure 15: Learning and regret bounds on a stochastic MDP.

actually it follows its bounds $\tilde{O}(S\sqrt{AT})$ [9] where $S$ is the number of states and $A$ is the number of actions.

For the example in Figure 14 we attempted to display our performance compared to several benchmark tabula rasa approaches to exploration. There are many other algorithms we could have considered, but for a short paper we chose to focus against the most common approach ($\epsilon$-greedy) the pre-eminent optimistic approach (UCRL2) and posterior sampling (PSRL).

Other common heuristic approaches, such as optimistic initialization for Q-learning can be tuned to work well on this domain, however the precise parameters are sensitive to the underlying MDP[6]. To make a general-purpose version of this heuristic essentially leads to optimistic algorithms. Since UCRL2 is originally designed for infinite-horizon MDPs, we use the natural adaptation of this algorithm, which has state of the art guarantees in finite horizon MDPs as well [4].

Figure 15a displays the empirical regret of these algorithms together with bootstrapped DQN on the example from Figure 14. It is somewhat disconcerting that UCRL2 appears to incur linear regret, but it is proven to satisfy near-optimal regret bounds. Actually, as we show in Figure 15b, the algorithm produces regret which scales very similarly to its established bounds [9]. Similarly, even for this tiny problem size, the recent analysis that proves a near optimal sample complexity in fixed horizon problems [4] only guarantees that we will have fewer than $10^{10}$ $\epsilon = 1$ suboptimal episodes. While these bounds may be acceptable in worst case $\tilde{O}(\cdot)$ scaling, they are not of much practical use.

### C.3 One-hot features

In Figure 16 we include the mean performance of bootstrapped DQN with one-hot feature encodings. We found that, using these features, bootstrapped DQN learned the optimal policy for most seeds, but was somewhat less robust than the thermometer encoding. Two out of ten seeds failed to learn the optimal policy within 2000 episodes, this is presented in Figure 16.

Figure 16: Bootstrapped DQN also performs well with one-hot features, but learning is less robust.

# D Experiments for Atari

## D.1 Experimental setup

We use the same 49 Atari games as [12] for our experiments. Each step of the agent corresponds to four steps of the emulator, where the same action is repeated, the reward values of the agents are clipped between -1 and 1 for stability. We evaluate our agents and report performance based upon the raw scores.

The convolutional part of the network used is identical to the one used in [12]. The input to the network is 4x84x84 tensor with a rescaled, grayscale version of the last four observations. The first convolutional (conv) layer has 32 filters of size 8 with a stride of 4. The second conv layer has 64 filters of size 4 with stride 2. The last conv layer has 64 filters of size 3. We split the network beyond the final layer into $K = 10$ distinct heads, each one is fully connected and identical to the single head of DQN [12]. This consists of a fully connected layer to 512 units followed by another fully connected layer to the Q-Values for each action. The fully connected layers all use Rectified Linear Units(ReLU) as a non-linearity. We normalize gradients $1/K$ that flow from each head.

We trained the networks with RMSProp with a momentum of 0.95 and a learning rate of 0.00025 as in [12]. The discount was set to $\gamma = 0.99$, the number of steps between target updates was set to $\tau = 10000$ steps. We trained the agents for a total of 50m steps per game, which corresponds to 200m frames. The agents were every 1m frames, for evaluation in bootstrapped DQN we use an ensemble voting policy. The experience replay contains the 1m most recent transitions. We update the network every 4 steps by randomly sampling a minibatch of 32 transitions from the replay buffer to use the exact same minibatch schedule as DQN. For training we used an $\epsilon$-greedy policy with $\epsilon$ being annealed linearly from 1 to 0.01 over the first 1m timesteps.

## D.2 Gradient normalization in bootstrap heads

Most literature in deep RL for Atari focuses on learning the best single evaluation policy, with particular attention to whether this above or below human performance [12]. This is unusual for the RL literature, which typically focuses upon cumulative or final performance.

Bootstrapped DQN makes significant improvements to the cumulative rewards of DQN on Atari, as we display in Figure 9, while the peak performance is much more We found that using bootstrapped DQN without gradient normalization on each head typically learned even faster than our implementation with rescaling $1/K$, but it was somewhat prone to premature and suboptimal convergence. We present an example of this phenomenon in Figure 17.

Figure 17: Normalization fights premature convergence.

We found that, in order to better the benchmark "best" policies reported by DQN, it was very helpful for us to use the gradient normalization. However, it is not entirely clear whether this represents an improvement for all settings. In Figures 18a and 18b we present the cumulative rewards of the same algorithms on Beam Rider.

(a) Normalization does not help cumulative rewards.

(b) Even over 200m frames the importance of exploration dominates the effects of an inferior final policy.

Figure 18: Planning, learning and exploration in RL.

Where an RL system is deployed to learn with real interactions, cumulative rewards present a better measure for performance. In these settings the benefits of gradient normalization are less clear. However, even with normalization $1/K$ bootstrapped DQN significantly outperforms DQN in terms of cumulative rewards. This is reflected most clearly in Figure 9 and Table 2.

## D.3    Sharing data in bootstrap heads

In this setting all network heads share all the data, so they are not actually a traditional bootstrap at all. This is different from the regression task in Section 2, where bootstrapped data was essential to obtain meaningful uncertainty estimates. We have several theories for why the networks maintain significant diversity even without data bootstrapping in this setting. We build upon the intuition of Section 5.2.

First, they all train on different target networks. This means that even when facing the same $(s, a, r, s')$ datapoint this can still lead to drastically different Q-value updates. Second, Atari is a deterministic environment, any transition observation is the unique correct datapoint for this setting. Third, the networks are deep and initialized from different random values so they will likely find quite diverse generalization even when they agree on given data. Finally, since all variants of DQN take many many frames to update their policy, it is likely that even using $p = 0.5$ they would still populate their replay memory with identical datapoints. This means using $p = 1$ to save on minibatch passes seems like a reasonable compromise and it doesn't seem to negatively affect performance too much in this setting. More research is needed to examine exactly where/when this data sharing is important.

## D.4    Results tables

In Table 1 the average score achieved by the agents during the most successful evaluation period, compared to human performance and a uniformly random policy. DQN is our implementation of DQN with the hyperparameters specified above, using the double Q-Learning update.[25]. We find that peak final performance is similar under bootstrapped DQN to previous benchmarks.

To compare the benefits of exploration via bootstrapped DQN we benchmark our performance against the most similar prior work on incentivizing exploration in Atari [20]. To do this, we compute the AUC-100 measure specified in this work. We present these results in Table 2 compare to their best performing strategy as well as their implementation of DQN. Importantly, bootstrapped DQN outperforms this prior work significantly.

|  | Random | Human | Bootstrapped DQN | DDQN | Nature |
|---|---|---|---|---|---|
| Alien | 227.8 | 7127.7 | 2436.6 | **4007.7** | 3069 |
| Amidar | 5.8 | 1719.5 | 1272.5 | **2138.3** | 739.5 |
| Assault | 222.4 | 742.0 | **8047.1** | 6997.9 | 3359 |
| Asterix | 210.0 | 8503.3 | **19713.2** | 17366.4 | 6012 |
| Asteroids | 719.1 | 47388.7 | 1032.0 | **1981.4** | 1629 |
| Atlantis | 12850.0 | 29028.1 | **994500.0** | 767850.0 | 85641 |
| Bank Heist | 14.2 | 753.1 | **1208.0** | 1109.0 | 429.7 |
| Battle Zone | 2360.0 | 37187.5 | **38666.7** | 34620.7 | 26300 |
| Beam Rider | 363.9 | 16926.5 | **23429.8** | 16650.7 | 6846 |
| Bowling | 23.1 | 160.7 | 60.2 | **77.9** | 42.4 |
| Boxing | 0.1 | 12.1 | **93.2** | 90.2 | 71.8 |
| Breakout | 1.7 | 30.5 | **855.0** | 437.0 | 401.2 |
| Centipede | 2090.9 | 12017.0 | 4553.5 | 4855.4 | **8309** |
| Chopper Command | 811.0 | 7387.8 | 4100.0 | 5019.0 | **6687** |
| Crazy Climber | 10780.5 | 35829.4 | **137925.9** | 137244.4 | 114103 |
| Demon Attack | 152.1 | 1971.0 | 82610.0 | **98450.0** | 9711 |
| Double Dunk | -18.6 | -16.4 | **3.0** | -1.8 | -18.1 |
| Enduro | 0.0 | 860.5 | **1591.0** | 1496.7 | 301.8 |
| Fishing Derby | -91.7 | -38.7 | **26.0** | 19.8 | -0.8 |
| Freeway | 0.0 | 29.6 | **33.9** | 33.4 | 30.3 |
| Frostbite | 65.2 | 4334.7 | 2181.4 | **2766.8** | 328.3 |
| Gopher | 257.6 | 2412.5 | **17438.4** | 13815.9 | 8520 |
| Gravitar | 173.0 | 3351.4 | 286.1 | **708.6** | 306.7 |
| Hero | 1027.0 | 30826.4 | **21021.3** | 20974.2 | 19950 |
| Ice Hockey | -11.2 | 0.9 | **-1.3** | -1.7 | -1.6 |
| Jamesbond | 29.0 | 302.8 | **1663.5** | 1120.2 | 576.7 |
| Kangaroo | 52.0 | 3035.0 | **14862.5** | 14717.6 | 6740 |
| Krull | 1598.0 | 2665.5 | 8627.9 | **9690.9** | 3805 |
| Kung Fu Master | 258.5 | 22736.3 | **36733.3** | 36365.7 | 23270 |
| Montezuma Revenge | 0.0 | 4753.3 | **100.0** | 0.0 | 0 |
| Ms Pacman | 307.3 | 6951.6 | 2983.3 | **3424.6** | 2311 |
| Name This Game | 2292.3 | 8049.0 | 11501.1 | **11744.4** | 7257 |
| Pong | -20.7 | 14.6 | **20.9** | **20.9** | 18.9 |
| Private Eye | 24.9 | 69571.3 | **1812.5** | 158.4 | 1788 |
| Qbert | 163.9 | 13455.0 | 15092.7 | **15209.7** | 10596 |
| Riverraid | 1338.5 | 17118.0 | 12845.0 | **14555.1** | 8316 |
| Road Runner | 11.5 | 7845.0 | **51500.0** | 49518.4 | 18257 |
| Robotank | 2.2 | 11.9 | 66.6 | **70.6** | 51.6 |
| Seaquest | 68.4 | 42054.7 | 9083.1 | **19183.9** | 5286 |
| Space Invaders | 148.0 | 1668.7 | 2893.0 | **4715.8** | 1976 |
| Star Gunner | 664.0 | 10250.0 | 55725.0 | **66091.2** | 57997 |
| Tennis | -23.8 | -8.3 | 0.0 | **11.8** | -2.5 |
| Time Pilot | 3568.0 | 5229.2 | 9079.4 | **10075.8** | 5947 |
| Tutankham | 11.4 | 167.6 | 214.8 | **268.0** | 186.7 |
| Up N Down | 533.4 | 11693.2 | **26231.0** | 19743.5 | 8456 |
| Venture | 0.0 | 1187.5 | 212.5 | 239.7 | **380** |
| Video Pinball | 0.0 | 17667.9 | **811610.0** | 685911.0 | 42684 |
| Wizard Of Wor | 563.5 | 4756.5 | 6804.7 | **7655.7** | 3393 |
| Zaxxon | 32.5 | 9173.3 | 11491.7 | **12947.6** | 4977 |

Table 1: Maximal evaluation Scores achieved by agents

We now compare our method against the results in [20]. In this paper they introduce a new measure of performance called AUC-100, which is something similar to normalized cumulative rewards up to 20 million frames. Table 2 displays the results for our reference DQN and bootstrapped DQN as Boot-DQN. We reproduce their reference results for DQN

as DQN* and their best performing algorithm, Dynamic AE. We also present bootstrapped DQN without head rescaling as Boot-DQN+.

|  | DQN* | Dynamic AE | DQN | Boot-DQN | Boot-DQN+ |
|---|---|---|---|---|---|
| Alien | 0.15 | 0.20 | 0.23 | 0.23 | **0.33** |
| Asteroids | 0.26 | 0.41 | 0.29 | 0.29 | **0.55** |
| Bank Heist | 0.07 | 0.15 | 0.06 | 0.09 | **0.77** |
| Beam Rider | 0.11 | 0.09 | 0.24 | 0.46 | **0.79** |
| Bowling | 0.96 | **1.49** | 0.24 | 0.56 | 0.54 |
| Breakout | 0.19 | 0.20 | 0.06 | 0.16 | **0.52** |
| Enduro | 0.52 | 0.49 | 1.68 | **1.85** | 1.72 |
| Freeway | 0.21 | 0.21 | 0.58 | 0.68 | **0.81** |
| Frostbite | 0.57 | 0.97 | 0.99 | **1.12** | 0.98 |
| Montezuma Revenge | 0.00 | 0.00 | 0.00 | 0.00 | 0.00 |
| Pong | 0.52 | 0.56 | -0.13 | 0.02 | **0.60** |
| Qbert | 0.15 | 0.10 | 0.13 | 0.16 | **0.24** |
| Seaquest | 0.16 | 0.17 | 0.18 | 0.23 | **0.44** |
| Space Invaders | 0.20 | 0.18 | 0.25 | 0.30 | **0.38** |
| **Average** | 0.29 | 0.37 | 0.35 | 0.41 | **0.62** |

Table 2: AUC-100 for different agents compared to [20]

We see that, on average, both bootstrapped DQN implementations outperform Dynamic AE, the best algorithm from previous work. The only game in which Dynamic AE produces best results is Bowling, but this difference in Bowling is dominated by the implementation of DQN* vs DQN. Bootstrapped DQN still gives over 100% improvement over its relevant DQN baseline. Overall it is clear that Boot-DQN+ (bootstrapped DQN without rescaling) performs best in terms of AUC-100 metric. Averaged across the 14 games it is over 50% better than the next best competitor, which is bootstrapped DQN with gradient normalization.

However, in terms of peak performance over 200m frames Boot-DQN generally reached higher scores. Boot-DQN+ sometimes plateaus early as in Figure 17. This highlights an important distinction between evaluation based on best learned policy versus cumulative rewards, as we discuss in Appendix D.2. Bootstrapped DQN displays the biggest improvements over DQN when doing well during learning is important.

## Footnotes

[6]Further, it is difficult to extend the idea of optimistic initialization with function generalization, especially for deep neural networks.