[Reviews · NeurIPS 2016]

Reviewer 1

Summary

The paper presents a new exploration strategy for deep reinforcement learning with discrete actions. The approach uses bootstrapped neural networks to approximate the posterior distribution of the Q-function. The bootstrapped network consists of several networks which use a shared layer for feature learning but separate output layers. Each network is learning from a slightly different data set, and hence, the different networks will learn slightly different Q-functions. The authors modify the DQN reinforcement learning algorithm to work with bootstrapped networks and show that Thompson sampling, i.e., sampling one q-function per episode, is a more effective exploration technique as the typically used "shallow" exploration. The algorithm is tested on a multitude of games in the Atari benchmark suite.

Qualitative Assessment

I really like this paper. Deep exploration is one of the key ingredients that are missing for effective reinforcement learning algorithms. The paper offers a quite elegant and effective solution for this problem, at least in discrete action domains. The paper is very well written and I really like the motivating example with the chain. The results also look convincing and they are quite exhaustive, as they include 49 Atari games. No complaints, good paper, accept.

Confidence in this Review

3-Expert (read the paper in detail, know the area, quite certain of my opinion)


Reviewer 2

Summary

In this paper, the authors propose a bootstrapped DQN method for incorporating exploration in reinforcement learning. The authors present experimental results on Arcade learning environment that shows favorable performance for some games, compared to existing method (DQN) without exploration.

Qualitative Assessment

Merits The idea of using bootstrapping, as a replacement of methods like posterior sampling, is interesting and promising. It has potential to be more robust to model misspecification, and in many cases easier to impement. Critique The paper is not very well written or organized. Proper background on Bootstrapping and DQN needs to be included before section 3, and Algorithm 1 should be included in Section 3. A number of observations have been laid out in the front of the reader without an overall compelling story. The point of Section 5 is not entirely clear to me. Do authors want to claimthat no other algorithm provides deep exploration? What is "Bandit algorithm" in Figure 2(a), what is "RL+shallow explore"? Is RL+deep explore same as bootstrapped DQN? Again, in Section 5.1, other algorithms need to be properly defined. What is an ensemble DQN? Authors mention "Thompson DQN is same as bootstrapped DQN but resamples every step" That doesn't seem the right version of Thompson Sampling if that is what authors are referring to by the name "Thompson DQN". Shouldn't TS be using posterior sampling? Or, if authors believe bootstrapping is equivalent to posterior sampling, such a result should be referred to. The captions of Figure 5(a) and 5(b) seem incorrect. And, finally the results in figure 6 don't seem to demonstrate very convincingly the benefits of bootstrapped DQN over DQN. Figure 7,8,9 looks better for some games.

Confidence in this Review

2-Confident (read it all; understood it all reasonably well)


Reviewer 3

Summary

First of all, let me say I am not an expert in DQN or neural networks, so this review may be a little superficial. The authors propose a modification of the DQN algorithm for reinforcement learning with neural network function approximation. The objective is to tackle the exploration problem, which is a big issue because the "default" epsilon-greedy approach has provable limitations in all but the simplest of setups. The description of the new approach is sometimes a little vague (see below), but the general approach is to construct several (K - a free parameter) neural networks (or one big neural network with K heads) to approximate the value function and play each episode by first randomly choosing a network and following its recommendations until the end of the episode and using the resulting data to train a random subset of the networks. The idea is to use the variation in the learning to simulate some kind of confidence bound so that the algorithm behaves a little bit like Thompson sampling.

Qualitative Assessment

The paper has some strengths and weaknesses. First of all, the experimental results are quite interesting, especially that the algorithm outperforms DQN on Atari. The results on the synthetic experiment are also interesting. I have three main concerns about the paper. 1. There is significant difficulty in reconstructing what is precisely going on. For example, in Figure 1, what exactly is a head? How many layers would it have? What is the "Frame"? I wish the paper would spend a lot more space explaining how exactly bootstrapped DQN operates (Appendix B cleared up a lot of my queries and I suggest this be moved into the main body). 2. The general approach involves partitioning (with some duplication) the samples between the heads with the idea that some heads will be optimistic and encouraging exploration. I think that's an interesting idea, but the setting where it is used is complicated. It would be useful if this was reduced to (say) a bandit setting without the neural network. The resulting algorithm will partition the data for each arm into K (possibly overlapping) sub-samples and use the empirical estimate from each partition at random in each step. This seems like it could be interesting, but I am worried that the partitioning will mean that a lot of data is essentially discarded when it comes to eliminating arms. Any thoughts on how much data efficiency is lost in simple settings? Can you prove regret guarantees in this setting? 3. The paper does an OK job at describing the experimental setup, but still it is complicated with a lot of engineering going on in the background. This presents two issues. First, it would take months to re-produce these experiments (besides the hardware requirements). Second, with such complicated algorithms it's hard to know what exactly is leading to the improvement. For this reason I find this kind of paper a little unscientific, but maybe this is how things have to be. I wonder, do the authors plan to release their code? Overall I think this is an interesting idea, but the authors have not convinced me that this is a principled approach. The experimental results do look promising, however, and I'm sure there would be interest in this paper at NIPS. I wish the paper was more concrete, and also that code/data/network initialisation can be released. For me it is borderline. Minor comments: * L156-166: I can barely understand this paragraph, although I think I know what you want to say. First of all, there /are/ bandit algorithms that plan to explore. Notably the Gittins strategy, which treats the evolution of the posterior for each arm as a Markov chain. Besides this, the figure is hard to understand. "Dashed lines indicate that the agent can plan ahead..." is too vague to be understood concretely. * L176: What is $x$? * L37: Might want to mention that these algorithms follow the sampled policy for awhile. * L81: Please give more details. The state-space is finite? Continuous? What about the actions? In what space does theta lie? I can guess the answers to all these questions, but why not be precise? * Can you say something about the computation required to implement the experiments? How long did the experiments take and on what kind of hardware? * Just before Appendix D.2. "For training we used an epsilon-greedy ..." What does this mean exactly? You have epsilon-greedy exploration on top of the proposed strategy?

Confidence in this Review

1-Less confident (might not have understood significant parts)


Reviewer 4

Summary

The paper tries to approach the problem of more efficient exploration in Deep RL using a distribution over Q-values. This distribution is implemented by random initialization of multiple policies over a common feature extractor. There are 2 experiments showcasing the estimation of a Gaussian process and more efficient and far-sighted exploration in a markov chain. The third experiment is on the Atari domain to show the scalability of this method. Particularly, the paper modifies the DQN architecture by adding multiple heads at the end of the network. It chooses heads to execute its policy at train time, giving directed exploratory actions. At test time, the multiple heads act as an ensemble, and provide a distribution over each action.

Qualitative Assessment

Using bootstrap methods to quantify uncertainty is a good idea and the paper certainly proves it through the experiments. The experimental methodology is good and showcases the capabilities of the technique well. The comparison with methods is appropriate for the most part. There is some curiosity if the recently proposed Asynchronous methods achieve a similar kind of exploration by sampling different parts of the state space. It would have been a worthwhile addition to compare these two methods. Since the different heads are learning from experience recorded from executing different policies, was this not sufficient to not require the experience replay? Was there a reason to keep the experience replay even if all the heads are learning from all the data? Most other implementation issues are considered in the appendix.

Confidence in this Review

3-Expert (read the paper in detail, know the area, quite certain of my opinion)


Reviewer 5

Summary

The paper proposes a different exploration method for deep RL, in order to speed up learning. The main innovation lies in changing the architecture of the DQN to incorporate multiple heads, each producing a Q-value for a different policy. Experiments demonstrate that the bootstrapped DQN is able to explore more efficiently and obtain higher rewards than the base DQN.

Qualitative Assessment

I enjoyed reading the paper and the experiments are convincing. A few comments: 1. Setting p=1 seems to work as well (or better) than other values. The authors provide some intuition in Appendix D, but I'm still not convinced completely. If the only difference between the heads lies in the initial values of the weights, and they each perform updates using the same transitions from experience replay, the policies learnt by each shouldn't be significantly different. If they are trained on the same loss values, shouldn't they approximately produce similar Q-values for the different (s, a) pairs? 2. The bootstrap DQN has more parameters than the DQN due to the multiple heads? Did you try comparing the two models with an equal number of parameters?

Confidence in this Review

3-Expert (read the paper in detail, know the area, quite certain of my opinion)


Reviewer 6

Summary

The purpose of this work is to introduce an efficient exploration strategy for DRL. For this goal, the authors suggest to learn multiple DQNs in parallel, while sharing some parameters, and demonstrate the efficiency of their method on different domains.

Qualitative Assessment

Given the current interest in DQN and its variants, this is definitely an interesting topic to investigate. The approach presented here is well motivated, straight forward to implement and computationally efficient. The paper is well written and easy to follow. The results are convincing and I think that researchers from the DRL community would benefit from reading this paper. My main criticism is that the paper mainly compares the bootstrapped DQN vs. the original DQN. First, I believe that the paper would benefit from comparing the proposed method with other exploration strategies. For example, other models for random value functions had been mentioned in the paper (lines 55-58) but were not demonstrated in practice and some variants (ensemble, Thompson) were considered in section 5.1 but not on the Atari domains. Second, the authors compare their results only with the vanilla DQN architecture and claim that the method taken in this paper is orthogonal to other improvements such as DDQN and the dueling network architecture. In my opinion, the fact that multiple but identical DQNs manage to solve the problem better than one is interesting, but it also implies that the single DQN still has inherent problems. It is not clear to me why the authors claim that their method is orthogonal to other methods, and actually it might help to solve some of the unstable behaviors of DQN. Therefore, I think that this work can benefit from combining it with other DQN variants. Small remarks: Line 179-180 what is the ensemble policy? Line 231 – DDQN -> DQN Appendix line 554 – broken ref.

Confidence in this Review

3-Expert (read the paper in detail, know the area, quite certain of my opinion)